# Improving Humanization through Metaverse-Related Technologies: A Systematic Review

**Maria Gonzalez-Moreno** [1,2]**, Paula Andrade-Pino** [1]**, Carlos Monfort-Vinuesa** [1,3,4]**, Antonio Piñas-Mesa** [5] **and Esther Rincon** [1,3,]*

1   Psycho-Technology Lab, Universidad San Pablo-CEU, CEU Universities, Urbanización Montepríncipe, 28660 Boadilla del Monte, Spain
2   Departamento de Ciencias Médicas Básicas, Facultad de Medicina, Universidad San Pablo-CEU, CEU Universities, Campus de Montepríncipe, Urbanización Montepríncipe, 28660 Boadilla del Monte, Spain
3   Departamento de Psicología y Pedagogía, Facultad de Medicina, Universidad San Pablo-CEU, CEU Universities, Urbanización Montepríncipe, 28660 Boadilla del Monte, Spain
4   Departamento de Medicina Interna, HM Hospitales, Universidad San Pablo-CEU, CEU Universities, Urbanización Montepríncipe, 28660 Boadilla del Monte, Spain
5   Departamento de Humanidades, Facultad Humanidades y CC Comunicación, Universidad San Pablo-CEU, CEU Universities, Paseo Juan XXIII 8, 28040 Madrid, Spain
*   Correspondence: maria.rinconfernande@ceu.es; Tel.: +34-913-724-700 (ext. 15076)

**Abstract:** While there is an increasing awareness regarding the culture of humanization, which is strongly needed in the healthcare environment, little knowledge has been provided in relation to accurate strategies to teach humanization skills to healthcare undergraduate students, as well as to healthcare professionals. Furthermore, the usefulness of new technologies to improve humanization skills has hardly been addressed so far in the scientific literature, including promising strategies such as Metaverse-related technologies. Presumably, this is the first systematic review focused on the efficacy of Metaverse-related technologies to increase the acquisition of humanization skills in the healthcare environment. The purpose of this study was to review the scientific studies published in the last decade to answer the following two questions: (1) are Metaverse-related technologies useful in enhancing humanization skills in the healthcare environment? (2) What are the advantages and disadvantages that should be addressed to successfully develop Metaverse-related technologies in the healthcare sector? We conducted a systematic review of the peer-reviewed literature from EBSCO, Ovid, PubMed, Scopus, and Web of Science (WOS), following the PRISMA statements and using the following keywords: "humanization + Metaverse"; "humanization + mixed reality"; "humanization + extended reality"; "humanization + augmented reality"; "humanization + virtual reality"; "humanization + app"; "humanization + telemedicine"; "humanization + digital health"; "humanization + eHealth"; "humanization + telehealth"; "humanization + web-based"; "humanization + website"; "humanization + digital"; "humanization + online"; and "humanization + internet". Studies published from 2012 to the present, written in the English language, were reviewed. A total of 505 records were obtained, of which three were selected based on the inclusion and exclusion criteria. The results will be helpful in developing new strategies to improve humanization skills in the health sphere.

**Keywords:** Metaverse; mixed reality; extended reality; humanization; education; undergraduates; students; healthcare

## 1. Introduction

Humanization entails "a process of communication and caring among people that leads to self-transformation and an understanding of the fundamental spirit of life and a sense of compassion and unity" [1], lending crucial relevance to the human rights approach to people [2]. Specifically, regarding the humanization of health, concepts such as the

visualization of the person in his or her integrality are encompassed, prioritizing his or her care needs while paying attention to his or her dimensions, i.e., social, spiritual, and relational [3]. Likewise, in a recent systematic review [4], it is stated that "respect for the patient's dignity, uniqueness, individuality and humanity" is a relevant skill to be taken into account in patient care.

Thus, specifically in the healthcare setting, the imperative need arises to integrate humanization or person-centered care, offering new psychosocial competencies (beyond technical competencies) of the various agents who interact with patients and their relatives, based on the training and specialization of the agents [5].

This aspect highlights the great challenge to both educational and health institutions in fostering competencies in this context among students, professionals, and agents who interact with patients, allowing for the incorporation of learning that enhances comprehensive care [6], in which the patient is considered in their entirety, i.e., considering them as a complete human being [7]. In this regard, in the person-centered care approach, special emphasis is placed on recognizing the beliefs, values, and preferences, as well as the individual needs, of the patient and his or her family [5,8]. This should play a leading role, together with the healthcare staff, to promote the quality of care and, therefore, the comprehensive health of the patient, differing transcendentally from the paternalistic healthcare vision [9].

However, education has undergone changes due to the incorporation of information and communications technology (ICT), which has driven new methods in the various teaching–learning processes [10]. In the health field, numerous benefits granted by ICTs are seen, not only in terms of care for patients but also regarding the promotion of academic growth processes in students in this field [11].

Within the use of ICT in various fields, extended or mixed reality (which integrates augmented and virtual reality) has proliferated, being a "technology with the ability to create and add information developed virtually with the knowledge and control of a real environment" [12]. In this regard, we find the Metaverse, which is "a next-generation Internet hypothesis consisting in a stable, decentralized, 3D virtual reality setting" [13], also described as "a digital universe accessible through a virtual setting" [14], linking "the physical world with a virtual environment" [15].

Regarding Metaverse technology, four categories are presented: augmented reality, lifelogging, the mirror world, and virtual reality [16]. Through the mirror world, it is possible to develop learning related to the real world, carrying it out in a virtual educational classroom; likewise, with virtual reality, it is possible to use avatars to interact and explore a virtual space [16].

Due to its characteristics, the Metaverse, in the context of education, facilitates the training and learning of various types of thinking and skills needed in the real world; it also enables problem-solving strategy development [16], as well as the development of improved communication, in which it is possible to adopt a more experiential approach, including the obtainment of a better understanding of emotions [17]. Likewise, through the exploration of a virtual world, favorable results are obtained in terms of learning, evidencing that the effective and expected changes of the users can be transferred to the real world, as can be observed from Serious Games, which have covered various areas of support, specifically in the health context, e.g., mental health and education, where multiple successful initiatives have been presented [18]. These initiatives, including the virtual pink dolphin (VPD) serious play, have explored learning and skills development support for children with special needs, which can be considered within their therapeutic scope [19]. Through virtual simulation, it is even possible to develop the learning of critical thinking, where favorable results have been found in terms of strengthening cognitive skills, as well as the satisfaction of students, who have presented greater self-confidence and greater acceptance of the teaching method [20]. On the other hand, training proposals are presented for teachers through digital learning under the e-learning system, favoring the opportunity to perform practices within the tasks considered, as well as the application of new ideas, actions, and experimentation, which can be complemented with the previous

learning of professionals and adapted to their work practices, favoring the delivery of knowledge to students from new professional practices that favor the stimulation and facilitation of students' learning [21].

Regarding the teaching of humanization in different educational areas, there is an opportunity to enhance the elements of humanization through the instrumentalization of ICT [22]. Thus, humanization skills could be promoted through extended reality (XR), specifically in the Metaverse, based on its immersive and interactive potential, which could benefit students through new ways of learning [14,23].

One of the challenges arising in the application of the Metaverse in diverse learning contexts is in identifying properly trained teams [14], in addition to the need for further research in the field of the application of the Metaverse in healthcare [17]. However, it should be noted that this potential is being exploited [24]. On the other hand, it poses numerous advantages, such as the recreation of a learning context in which it is possible to simulate diverse settings—the field of education in this case—thus creating a space that fosters exposure to numerous issues, practice in resolving them, and/or encouragement in the acquisition of specific skills and learning outcomes, which resembles reality, where avatars make it possible to replace issues that could arise in real life, allowing for students to practice/utilize what they have learned in relation to healthcare [17,25].

To this end, a meta-analysis (N = 32) was conducted to examine whether there were differences in the degree of social influence between two types of virtual depictions that can be used in the Metaverse: specifically, avatars and agents [26]. These two differ in that avatars are controlled by humans, whereas agents are controlled by computer algorithms. The results showed that, specifically in healthcare settings, following a clinical diagnosis, patients are more likely to comply with the advice given by avatars than by agents, resulting in avatars being more effective than the latter. In other words, avatars can wield greater social influence on individuals than agents, but agents are cheaper than avatars, so perhaps hybrid solutions could produce successful results as teachers, showing that such hybrid solutions (a teachable agent) invoke a greater sense of responsibility that motivates students' learning [27]. Furthermore, the perception of human control in these interactions—not merely a human-like appearance—was highly recommended.

The purpose of this study was to review the scientific studies published in the last decade to answer the two following questions: (1) Are Metaverse-related technologies useful in enhancing humanization skills in the healthcare environment? (2) What are the advantages and disadvantages that should be addressed to successfully develop Metaverse-related technologies in the healthcare sector?

## 2. Materials and Methods

### 2.1. General Description

A systematic search strategy was applied in November 2022 to detect all the relevant studies involving the use of Metaverse-related technologies to enhance humanization skills acquisition in the healthcare environment. It was performed and reported using the Preferred Reporting Items for Systematic Reviews and Meta-Analyses (PRISMA) Statement (see study protocol in File S1) [28]. The protocol was registered with the PROSPERO International Prospective Register of Systematic Reviews (CRD42023399388).

### 2.2. Selection Criteria

Inclusion criteria: The study papers, which were journal articles involving Metaverse-related technologies to increase humanization skills acquisition in the healthcare environment, were considered relevant. Studies had to be published in the English or Spanish language during the last decade (between 2012 and November 2022), providing specific outcomes (quantitative results).

Exclusion criteria: Studies that did not involve humanization training or did not include Metaverse-related technologies (e.g., a digital newsletter) were discarded. Protocols with unpublished results, narrative reviews, no journal articles (conference proceedings,

book chapters, or thesis), or published in a language other than English or Spanish were also excluded.

### 2.3. Outcomes

The primary outcomes were the type of training using Metaverse-related technologies and its usefulness in enhancing humanization skills. The secondary outcomes were the main advantages and disadvantages of the training developed, as well as the students' satisfaction levels with the Metaverse-related technologies.

### 2.4. Search Methodology

A comprehensive search was carried out in EBSCO (Academic Search Complete, CINAHL Plus with Full Text, Communication Source, eBook Collection, E-Journals, ERIC, Fuente Academica Premier, Humanities International Complete, MEDLINE, MLA Directory of Periodicals, MLA International Bibliography, OpenDissertations, PSICODOC, Psychology and Behavioral Sciences Collection, PsycInfo), Ovid, PubMed, Scopus, and WOS (Web of Science Core Collection), from inception until November 2022. The detailed search strategies used in all databases are provided in the File S1. All original research articles were retrieved for examination, and a search library was created using RefWorks©, a bibliography management program.

### 2.5. Data Collection and Analysis

Two authors (MG and ER) independently evaluated and reviewed all titles and abstracts thoroughly, focusing on the following three aspects. First, each title was assessed, and then each abstract, and lastly, each paper that contained a relevant title and abstract, according to the reviewer, was extracted. Following this, the inter-rater agreement between the two investigators (MG and ER) was obtained by calculating the Cohen kappa scores. SPSS version 27 (IBM Corp) was used to inform the interpretation of the Cohen kappa coefficient and was based on the categories developed by Douglas Altman [29] as 0.00–0.20 (poor), 0.21–0.40 (fair), 0.41–0.60 (moderate), 0.61–0.80 (good), and 0.81–1.00 (very good). If any inconsistency was found, a third author was consulted (CM). Cross-checking was carried out to identify any errors or oversights (ER). Any other discrepancies were fixed by the main team, only turning to the broader research team when necessary.

### 2.6. Data Extraction and Management

We extracted data based on (1) publication year, (2) country, (3) study design, (4) study aim, (5) sample size and mean participants' age, (6) targeted participants, (7) training using Metaverse-related technologies, (8) usefulness to enhance humanization skills, (9) main advantages/disadvantages, and (10) participants' satisfaction.

### 2.7. Quality of Included Studies

Given the variety of the research designs, the quality of the included studies was appraised using the Mixed Methods Appraisal Tool (MMAT), developed in 2006 [30] and revised in 2018 [31]. The highest values indicated the lower quality of the included studies (see File S2). One author (CM) independently extracted data on outcomes from all studies. The data was reviewed thoroughly by one reviewer (ER).

### 2.8. Statistical Analysis

Data were pooled using the program SPSS v. 27 (IBM Corp), which allowed for an analysis of frequencies (percentages) as well as means.

## 3. Results

### 3.1. Study Selection and Inclusion

By the use of an electronic database search, a total of 505 records were included in RefWorks©. From those, 274 duplicates were removed, and another 90 studies were

discarded because they did not meet the inclusion criteria (no journal papers). Then, 141 records were evaluated based on the titles and abstracts. Of these, 135 were eliminated because they did not comply with the inclusion criteria. Accordingly, the six remaining papers were selected for a full-text reading; of those six papers, three [32–34] were eliminated for different reasons (see File S3). A total of three publications were finally included [35–37]. Cohen's kappa indicated a significant level of agreement, and it was categorized as "good" ($\kappa = 0.70$) (range 0.61–0.80) based on the categories developed by Altman [16]. The PRISMA flow diagram [28] is shown in Figure 1. All the final studies were proven to be of sufficient quality to contribute equally to the thematic synthesis.

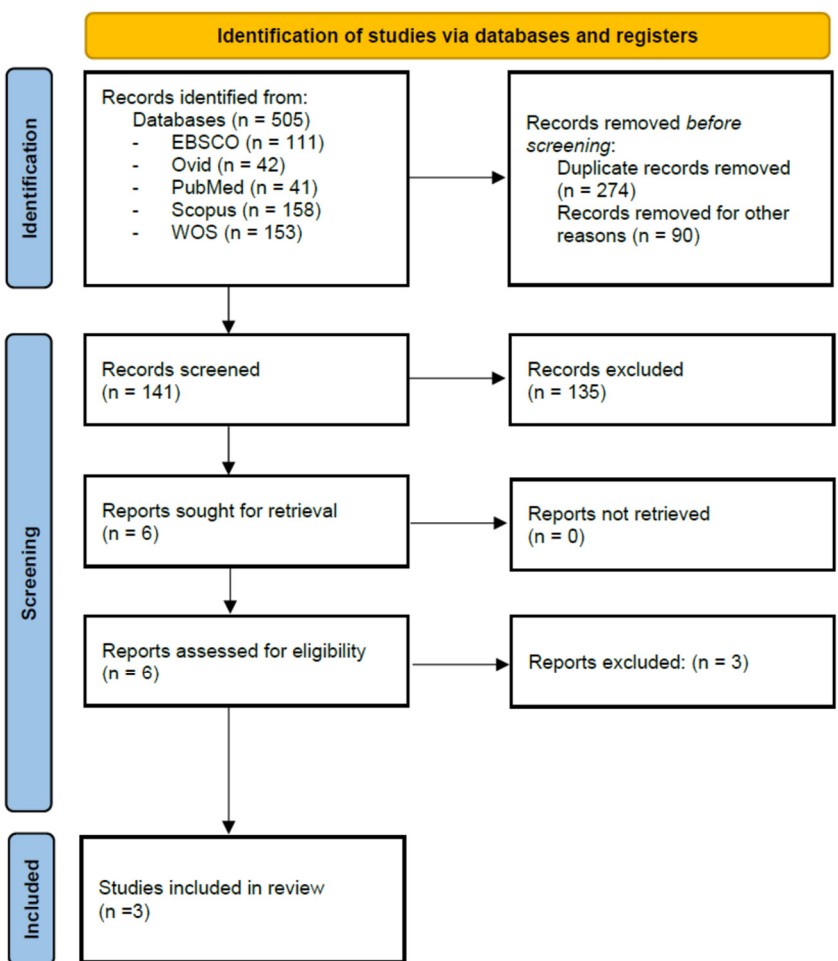

**Figure 1.** Systematic review of the literature flowchart.

*3.2. General Characteristics of the Studies Included*

Concerning points 1 (year of publication), 2 (country of the study), and 3 (study design), the following results were obtained (Table 1): the three studies that were chosen were published between 2021 ($n = 1$) [37] and 2022 (66.66%) ($n = 2$) [35,36]. The studies were conducted in Chile (33.33%) [35], Sweden (33.33%) [36], and Spain (33.33%) [37].

**Table 1.** General characteristics of included studies ($n = 3$).

| Study | Publication Year | Country | Study Design |
|---|---|---|---|
| Castillo-Parra et al. [35] | 2022 | Chile | Qualitative |
| Hallqvist [36] | 2022 | Sweden | Qualitative |
| Jiménez-Rodríguez et al. [37] | 2021 | Spain | Mixed Methods |

The studies involved followed a qualitative (*n* = 2; 66.66%) [35,36] or mixed-methods approach (*n* = 1; 33.4%) [37] (Table 1).

Addressing points 4 (study aim), 5 (sample size and participants' mean age), and 6 (targeted participant), the following results were produced. The objectives of the studies were immensely different (Table 2). In 33.4% (*n* = 1) of the studies, Metaverse-related technologies were used to explore how a digital caregiver is humanized through the health-enhancing approaches of personalization and friendliness for use in older patients [36]. Other goals included using Metaverse-related technologies to describe the experience of implementing the online training entitled "Humanization of the training processes in nursing, care for all", as well as to analyze their interventions in the online forum (*n* = 1) [35], or to examine the effects of virtual simulation-based training on the development and cultivation of humanization abilities in undergraduate nursing students (*n* = 1) (Table 2).

**Table 2.** General characteristics of studies included (II) (*n* = 3).

| Study | Study Aim | Sample Size (Mean Age) | Targeted Participants |
|---|---|---|---|
| Castillo-Parra et al. [35] | To describe the experience of implementing the online training labeled "Humanization of the training processes in nursing, care for all", as well as to analyze their interventions in the online forum. | 12 (Not provided) | Professors from a nursing school |
| Hallqvist [36] | To examine how a digital caregiver is humanized through the health-enhancing approaches of personalization and friendliness for use in older patients. | Not provided/Not provided | Researchers from computing science fields and researchers within occupational therapy and nursing fields |
| Jiménez-Rodríguez et al. [37] | To examine the effects of virtual simulation-based training on developing and cultivating humanization abilities in undergraduate nursing students. | 60 (23.83) | 3rd-year students, degree in nursing |

### 3.3. Assessment of the Methodological Quality of the Studies Included

Considerable diversity was found in the design of the studies, as well as in the statistical methods used, with heterogeneity in the presentation of the obtained results (see File S2).

### 3.4. Primary Outcomes

In relation to points 7 (training with Metaverse-related technologies) and 8 (useful in enhancing humanization skills), the following conclusions were established (Table 3).

The training sessions were used to address the following goals. In 33.4% (*n* = 1) of the studies, six case studies concerned virtual training, supported by the reading of texts, reviews of web pages, audiovisual materials, and simulated situations, encouraging debates and observations in the forum [35]. In another case (33.4%), different types of events lasting a total of 50 h, such as a user study with two researchers and two user study participants, meetings, seminars, public events (lectures, theme days), and social events, were observed, finding personalization and friendliness to be central health-enhancing practices in the humanization of the digital caregiver [36]. In the last case, six simulated scenarios focusing on basic healthcare at patients´ homes were developed by a virtual platform of online video conferences provided by the university (Blackboard Collaborate Launcher TM) [37].

**Table 3.** Primary outcomes (*n* = 3).

| Study | Training Using Metaverse-Related Technologies | Useful in Enhancing Humanization Skills |
|---|---|---|
| Castillo-Parra et al. [35] | Six case studies concern virtual training, supported by the reading of texts, reviews of web pages, audiovisual materials, and simulated situations, encouraging debate and observations in the forum. | Not provided |
| Hallqvist [36] | Different types of events, lasting a total of 50 h, such as a user study with two researchers and two user study participants, meetings, seminars, public events (lectures, theme days), and social events. | Not provided |
| Jiménez-Rodríguez et al. [37] | A virtual platform of online video conferences provided by the university (Blackboard Collaborate Launcher TM) was used to develop six simulated scenarios focusing on basic healthcare at patients' homes. | Not provided |

*3.5. Secondary Outcomes*

In regard to points 9 (main advantages and disadvantages) and 10 (participants' satisfaction), the subsequent results were obtained (Table 4).

**Table 4.** Secondary outcomes (*n* = 3).

| Study | Main Advantages and Disadvantages | Student Satisfaction |
|---|---|---|
| Castillo-Parra et al. [35] | Yes | Not provided |
| Hallqvist [36] | Yes | Not provided |
| Jiménez-Rodríguez et al. [37] | Yes | Not provided |

Among the advantages mentioned, Metaverse-related technologies offered a method for creating successful training in which the participants' reflections allowed us to understand conceptual and experiential elements of the humanization of care and of training. The participants disclosed the aspects in which they could exert an influence in achieving a more humanized culture, such as self-recognition and the acknowledgment of their students as individuals in a particular context [35]. In another case [36], the use of validated scales comprised humanization abilities, such as self-efficacy or empathy.

However, the main limitation of this study [36] has to do with the specific disadvantage which occurs in both simulated and real-life nursing video consultations: technical issues. Providing adequate network access and the proper functioning of virtual platforms could reduce this potential problem. In addition, another [36] limitation found was that humanization teaching entails that educators must view their students as whole individuals, taking into account the human aspects that affect their learning processes as future professionals: communication skills, problem-solving, and teamwork.

Finally, the research exploring the humanization of a digital caregiver in the Metaverse obtained through the health-enhancing practices of personalization and friendliness for use in older patients' homes to increase their health [36] shows that the personalization and friendliness of digital caregivers can serve as a means of providing personalized healthcare, while, at the same time, it may also result in a risk of patients believing that the digital caregiver is only supposed to follow the patient's health-related preferences and that the digital caregiver is a compliant friend.

A discursive conflict was identified between the patient discourse of self-determination and the healthcare professional discourse of authority and medical responsibility.

**4. Discussion**

The use of Metaverse-related technologies to teach humanization skills remains a broadly unexplored research topic. For this reason, the present study goal was to answer

two key questions: (1) whether Metaverse-related technologies could be considered useful tools in enhancing humanization skills in the healthcare environment, and (2) which advantages and disadvantages should be addressed to successfully develop Metaverse-related strategies in healthcare education.

*4.1. The Usefulness of Metaverse-Related Technologies in Enhancing Humanization Skills*

There is currently no standardized or normalized description of the concept of humanization applied to improve the development of certain professionals, such as healthcare professionals. However, recently, there has been an enhanced amount of research into the humanization of healthcare, as well as the creation of institutes, chairs, and academic initiatives aiming to improve the humanization of healthcare practices. Tools have also been designed to evaluate the "humanization index" of healthcare centers.

The term humanization, in the context of ethics, is used to denote, for example, the practice of a professional by assessing whether or not he or she puts into practice values such as respect for the dignity of patients and their families [38], autonomy, and confidentiality. In addition, psychological competencies, such as empathy and communication skills, which aim to improve the treatment of a patient in a vulnerable situation, are observed in a professional who carries out his or her work in a humanized way.

The objective of resources aimed at the humanization of health is to improve the ethical and psychological competencies of the professional, as well as the scenario in which the health practice takes place. However, these measures would also include improving the relationship between patients and their families and health professionals (Figure 2).

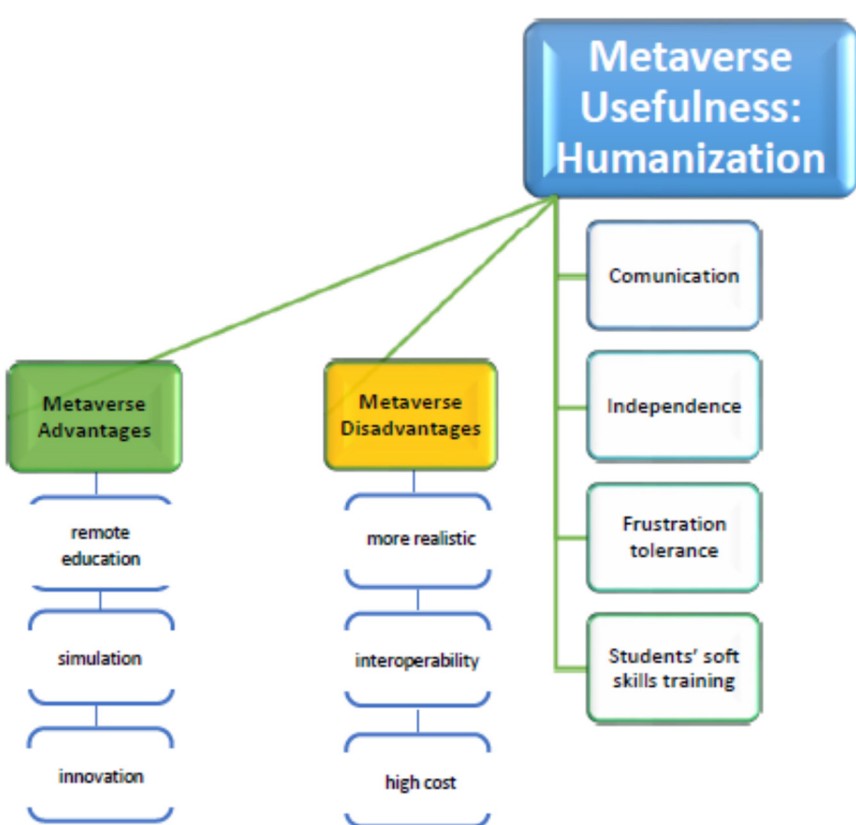

**Figure 2.** Advantages and disadvantages of Metaverse-related technologies in health education Metaverse.

Thus, the term "humanization", both in the cited literature and in other similar sources, adopts an ethical sense: human beings, in their different relationships, both personal and work-related, act in a humanized or dehumanized way and, at the same time, in relation to

these actions, they humanize or dehumanize themselves and, consequently, humanize or dehumanize the scenarios in which they act.

The training of competence in humanization implies both theoretical and practical training, with current immersive tools such as the Metaverse being a possibility for the improvement of these teachings.

Video games are presented as a potential tool to create specific metaverses that can help prepare students with soft skills. Among them are communication and creativity. The ability to interact with familiar and unknown people is arguably one of the most fundamental abilities to learn in different situations. This skill and others, such as independence and frustration tolerance, could be taught effectively by games that involve communication [39].

One recent study endeavored to improve university students' soft skills by using a humanization curriculum [40]. Specifically, the authors developed an instructional model, called the CELER Model, to enhance comprehensive humanization in students with a higher education, which consisted of the following steps: (1) the creation of an atmosphere (C), (2) an experience review (E), (3) learning for living (L), (4) empowerment (E), and (5) reflection (R). The students demonstrated that CELER was an effective strategy for the development of soft skills, such as strengthening positive attitudes, understanding oneself and others, working with others well, and creative thinking and problem-solving in problematic situations. All of these are crucial characteristics that must be trained and achieved among healthcare professionals.

### 4.2. Advantages and Disadvantages of Metaverse-Related Technologies in Health Education

The Metaverse affords some benefits to healthcare practitioners (Figure 2), such as being able to care for patients via online service. As a result, the implementation of Metaverse techniques with healthcare patients may lead to reduced costs for patients as well as for healthcare providers while also improving the healthcare given. Furthermore, the Metaverse allows for all patients and clinicians to converge in the same reality (virtual or physical) to be able to receive adequate treatment or to deliberate the best treatment that could be provided to a specific patient. The possibility of monitoring patients in real-time should also be mentioned because it facilitates the tracking of factors such as compliance, which could also be enhanced by using wearables. For instance, if patients were to use various types of body sensors (wearables) along with Metaverse-related technology (such as headsets for virtual reality or mixed reality), they would be able to control their own avatars with their whole bodies.

The Metaverse is being explored as one of the tools that can change, improve, and possibly transform healthcare in the future. The five areas that it could affect are collaborative work, education, clinical care, wellness, and monetization [41]. A metaverse of "medical technology and AI" (MeTAI) can facilitate the development, prototyping, evaluation, regulation, translation, and refinement of AI-based medical practice, especially medical image-guided diagnosis and therapy [42].

The field of education could also benefit from Metaverse-related technology, as undergraduate students could profit from remote education taking place in the Metaverse in an asynchronous manner, benefiting from this form of additional training.

In the educational field, the use of AR (augmented reality) and VR (virtual reality) will change the way that we learn and train in the medical field, as well as various processes and procedures. Virtual reality allows students to literally enter the human body; it allows the visualization of the body and the replication of real procedures. AR also provides students with hands-on learning, such as simulating patient and surgical encounters, allowing medical students to visualize and practice new techniques [41].

Given that the use of new technology during simulations is an enticing innovation for many students, which enhances their learning experience and perceived self-efficacy [43], their engagement [43,44], their enjoyment [45], their user satisfaction level [43,46,47], and the quality of the teaching and learning process [48], MR could be highly useful as a training tool for undergraduate mental health students by increasing their willingness to

acquire real knowledge [49] and bolstering their practical skills [46,48,50] compared with traditional teaching models [46], consequently improving their final performance [48].

Another advantage that Metaverse-related technologies could offer to healthcare educators is the ability to develop "health digital twins" (HDT), meaning "a virtual representation (digital twin) of a patient (physical twin) that is generated from multimodal patient data, population data and real-time updates on patient and environmental variables" [51]. The use of these virtual depictions or patients' avatars has already been shown to be useful in treating some chronic patients [52] by diminishing anxiety and depression symptoms and increasing variables such as quality of life, knowledge, and self-care behavior.

One of the Metaverse-related technology barriers mentioned in the studies included involves training participants to reflect personally on concepts and experiences that make it possible to discuss the humanization of care and also about their own experiences of caring for others and being cared for. They showed a positive view of how, after these virtual training sessions, their perception of emotional understanding and empathy had improved.

It is interesting to observe how Metaverse-related technologies are used to humanize technology that will be helpful for older patients. In this case, two different personalization practices were applied: selecting interfaces in which the avatars more closely resembled people and assigning personality traits and characteristics to the digital caregiver, which the patient recognized as friendliness [36]. This is similar to the empathy that health science professors and students aim to be taught.

The issue addressed in this project, however, exposes the difficulties in developing this type of setting: how to achieve a balance between obliging empathy, in which all the elderly patient's requests would be fulfilled, and persuasive empathy, which considers the emotional needs of elderly patients but prioritizes their health [36].

In order to achieve a balance between accommodating empathy and persuasive empathy, it is necessary to understand both the environment in which the interaction that will lead to an empathic response takes place and the co-ordination between the people who will exercise this empathic behavior. Most researchers (at least in the field of cognitive neuroscience and psychology) agree that empathy entails the adoption of the affective state of another person so that both the empathic subject and the target of this empathy are in a similar state [53].

According to Hoffmann's moral socialization theory, empathy involves cognitively interpreting what is happening to another person but also situating it in a way that is appropriate to the situation that is unfolding [54]. From this perspective, it is possible to understand the affective part of the empathic response ("feeling congruent with what the other person feels") and the cognitive part ("knowing what the other person feels") [55]. Identifying the emotions of the other person does not necessarily imply granting all their wishes or needs (complacent–affective empathy), nor does it necessarily imply only prioritizing health and relegating important emotional issues to the background (persuasive–cognitive empathy), so which variable can bring the two concepts together? It is at least necessary to obtain a balance between the two.

This is where the moral responsibility of the caregiver is important. In this regard, [56] there is a moderate positive correlation between cognitive empathy and moral sensitivity and a lower correlation between moral sensitivity and affective empathy. In order to be empathetic in the exercise of caring for others, health professionals need to display both types of empathy (cognitive and affective), as well as moral sensitivity, to be able to handle the distress and suffering of the people whom they care for. When these individual variables are aligned, they promote effective and humanized healthcare [57], thereby achieving behavior that satisfies both the primary task of the caregiver and the needs of the patient. It is from this perspective that the term "responsible empathy" can be coined, as empathic behavior implies a benefit for both parties (Figure 3).

| FLOW REALITY VIRTUALITY | | | | |
|---|---|---|---|---|
| | REAL ENVIRONMENT | AUGMENTED REALITY | INCREASED VIRTUALITY | VIRTUALREALITY |
| **INTERACTION LEVELS** | physical world | digital data and images in the real world | immersive virtual environments: VIDEOGAMES | more immersive virtual environments. METAVERSE |
| **NARRATIVE EMBEDDED IN THE PROCESS. BELONGS TO THE STORY ITSELF** | physical environment | reality 3D/scrip | | digital narrative space/script |
| **EMERGING NARRATIVE IN THE PROCESS. THE NARRATIVE THAT EMERGES AS VIRTUAL INTERACTION TAKES PLACE** | interactions and actions in physical space | actions generate information emergence in mixed environments | | actions based on interaction with the virtual environment |
| **EMPATHY** | cognitive empathy | emotional empathy | Responsible empathy | Embody Virtual Reality empathy |

**Figure 3.** Flow in virtual reality.

In the above diagram (Figure 3), it is represented the different levels of virtual interaction, the form of communication, and the empathy implicit in these levels, following the adaptation of Rubio-Tamayo et al. (2016) [58]. The diagram shows the different interaction and cognitive levels of the real and virtual worlds in the taxonomic classification originally carried out by Milgram and Kishino (1994) [59]. This taxonomy presents the idea that, at some point in the flow between these real and virtual worlds, there is a merger. There is a "continuum of virtuality" that connects completely real environments with completely virtual environments [58].

On this continuum, it can be superimposed a taxonomy of different forms of communication in which the features would be in accordance with Churchill et al. (2012) [60]: the environment is a collective space. This allows for the acquisition of awareness of the existence of others. This, in turn, allows for fluid negotiation and communication dynamics and presents multiple points of view [58].

Virtual reality is projected as a system of interaction with diverse levels of immersion, encompassing several sensory dimensions [59]. The overlapping forms of communication represent, in the "continuum of virtuality", the dimensions of human perception, interaction with the environment, and cognition, as well as the construction of stories in the style of earlier references, such as cinema or literature. These forms of communication are common in the training of students in health care because of their potential to improve soft skills and the construction of very important conceptual elements in the framework of the medical humanities [61–63], so they could be useful as a first approximation of how soft skills could be trained in the Metaverse.

Rubio-Tamayo et al. (2016) [58] also offer some examples of how these technologies offer increased empathetic experiences thanks to the possibility of direct immersion in a concrete environment. A relevant example is the virtual reality initiative developed by the United Nations High Commissioner for Refugees, UNHCR, to explain the plight of Syrian refugees: Clouds over Sidra, filmed in 2015 in the Zaatari refugee camp. This marked the beginning of a series of campaigns with this technology [64]. There is not yet a relevant number of comparative studies focused on the degree of empathy with the elements that

compose the immersive world in virtual reality, but many of the applications implicitly entail an increase in the empathy factor [58].

Of the different ways in which empathy can be trained in the virtual reality world, perhaps the closest to the Metaverse experience is the so-called embodied virtual reality (EVR) [65], a virtual reality technique that creates the illusion of being in another person's body [66,67]. With virtual reality, researchers apply motor and multisensory stimuli in synchrony with the first-person perspective of an avatar using computer-generated imagery [67] or the imagery of real humans through stereoscopic video [66]. In these studies, evidence shows that subjects feel as if they have exchanged bodies with another person [66]. Experiences of EVR allow users to essentially relate to others and observe the world from their perspectives [65].

One of the main disadvantages of Metaverse-related technologies is that it is not affordable to all individuals in society, so this technology is limited to those universities, educational institutions, and hospitals that have sufficient resources. Furthermore, as the achievement of appropriate interoperability is a critical goal of today's healthcare paradigm, further research should be conducted to fulfill this requirement, as well as to achieve more realistic avatars, which could lead to interactions that are perceived as more real [68], considering the previously reported benefits and depending on the type of avatar used [52].

Further research is needed to validate existing simulators and to verify whether improvements in performance in a simulated scenario translate into improved performance in real patients [69].

## 5. Conclusions

In accordance with the objectives proposed and the results of this systematic review, the following conclusions may be reached.

(1) There is not sufficient empirical evidence to be able to confirm that Metaverse-related technologies are an effective type of technology to increase the acquisition of humanization skills in the healthcare sphere; result k of these studies focused on this topic;

(2) Taking into account the different MR techniques developed with participants, numerous strengths and weaknesses should be addressed in order to successfully develop Metaverse-related strategies in healthcare education.

It is crucial to develop accurate studies that allow for a clear understanding of the drivers and barriers to successfully developing Metaverse-related technologies to improve humanization in healthcare education. In this sense, many remaining questions need to be addressed by future randomized controlled trials. Some of them could be mentioned as the following: does the effectiveness of Metaverse-related technologies depend on the type of participants involved (e.g., students, healthcare staff, or academics)? Are there specific modalities of Metaverse-related strategies that may produce greater enhancement, engagement, and acquisition regarding humanization abilities in the healthcare domain?

Due to the extreme scarcity of research, further studies are needed because their results could improve not only the students' skills but also the quality of the healthcare services provided to mental health patients.

### Limitations

The scarcity of studies and research based on Metaverse-related technologies in improving humanization skills in the healthcare environment must be highlighted. The diversity of included studies, as well as the scarcity of RCTs, made strong empirical evidence difficult to achieve. Likewise, the devices used for the Metaverse are diverse and their different modes of implementation caused the methodologies to vary, which prevented us from unifying and standardizing the results.

**Supplementary Materials:** The following supporting information can be downloaded at: https://www.mdpi.com/article/10.3390/electronics12071727/s1. File S1: Study protocol. File S2: Quality assessment of included studies (MMAT). File S3: Reasons for study exclusion.

**Author Contributions:** E.R. led the conception and design of the study, screening of included studies, data analysis and interpretation and wrote the first draft of the manuscript. M.G.-M. and E.R. were responsible for data extraction. M.G.-M., P.A.-P., C.M.-V., A.P.-M. and E.R. substantially contributed to the analysis and data interpretation and revised the work critically. All authors have read and agreed to the published version of the manuscript.

**Funding:** This work was funded by the grant "MPFI20AP" offered at Universidad San Pablo-CEU, CEU Universities (Madrid, Spain).

**Conflicts of Interest:** The authors declare no conflict of interest.

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
