# Peer review of "Improving Humanization through Metaverse-Related Technologies: A Systematic Review"

_electronics, doi:10.3390/electronics12071727_

Round 1

Reviewer 1 Report

In a general sense, the scope of the paper's objective is somewhat extensive within the context of the "Systematic review." Although the paper was badly written, I can see that it makes an attempt to cover a wide range of "Metaverse-related technologies for strengthening humanization skills," however it falls short in the following areas:

1.       About 80% of the content of the paper lies within the PRISMA systematic literature review without regards to the substance of Metaverse-related technologies for enhancing humanization skills. That is without respect to the substance of Metaverse-related technologies for improving humanization abilities, approximately 80% of the material of the paper is located within the PRISMA systematic literature review.

2.       There is no foundational breakdown of the presentation scheme of Metaverse-related technologies for enhancing humanization skills nor does it proposed a simplified way of understanding Metaverse-related technologies for enhancing humanization skills. I mean the paper does not suggest a simplified manner of understanding Metaverse-related technologies for boosting humanization abilities, nor does it provide a foundational breakdown of the presentation scheme of Metaverse-related technologies for enhancing humanization skills.

3.       Despite the fact that this is a "Systematic Review Paper," there is still a requirement to highlight an architectural flow on how Metaverse-related technologies can improve humanization skills.

4.       When you refer to "humanization," there is neither a conceptual nor a practical description of what you mean by this term. I mean there is no concept or operational definition of what you mean by Humanization.

5.       SECTION 4, which is supposedly where you should have outlined your review inference, but unfortunately you left it empty; if you had broadly explained "how to find a balance between obliging empathy and persuasive empathy" in another concept, that would have been a great contribution to the work; unfortunately, you did not do so.

Author Response

Reviewer 1:

Comments and Suggestions for Authors

In a general sense, the scope of the paper's objective is somewhat extensive within the context of the "Systematic review." Although the paper was badly written, I can see that it makes an attempt to cover a wide range of "Metaverse-related technologies for strengthening humanization skills," however it falls short in the following areas:

We really appreciate this suggestion. We will submit the paper for MDPI English editing service, and, hopefully, it will clarify the written message.

And thank you so much for taking the time to revise this manuscript, and consequently, helping us for its significant improvement.

Please see below the point-by-point response to each comment suggested.

  1. About 80% of the content of the paper lies within the PRISMA systematic literature review without regards to the substance of Metaverse-related technologies for enhancing humanization skills. That is without respect to the substance of Metaverse-related technologies for improving humanization abilities, approximately 80% of the material of the paper is located within the PRISMA systematic literature review.

Thank you very much for pointed out this improvement. Indeed, in our original text we had given great importance to the systematic review of the PRISMA literature against Metaverse-related technologies to enhance humanization skills. To improve this imbalance, we have expanded the introduction in this regard, as follows:

We have included this information in Introduction section (page 2, lines 72-76) explaining those changes undergone in education because of the incorporation of Information and Communication Technologies (ICT), which have promoted new methods in the different teaching-learning processes with special attention to the academic growth processes in healthcare students.

(Page 2, lines 77-83) we refer to the proliferation of extended reality, both augmented reality and virtual reality, and their capacity to create and add information from a real environment.

(Page 2-3, lines 84-93) we define Metaverse and what categories it presents and the characteristics that make it a training and learning tool in the context of education.

Also, in Discussion section, we have reviewed the literature on Metaverse as follows:

lines 361-375 (page 10) a thorough explanation of humanization and soft skills, especially in the context of medical education.

In lines 377-379 (pages 10-11) different possibilities of virtual reality are presented, including in Metaverse related to soft skills education.

In lines 402-407 (page 11) different applications of Metaverse in medical practice are presented.

In lines 409- 418 (page 11) as the field of education could also benefit from Metaverse-related technology.

  1. There is no foundational breakdown of the presentation scheme of Metaverse-related technologies for enhancing humanization skills nor does it proposed a simplified way of understanding Metaverse-related technologies for enhancing humanization skills. I mean the paper does not suggest a simplified manner of understanding Metaverse-related technologies for boosting humanization abilities, nor does it provide a foundational breakdown of the presentation scheme of Metaverse-related technologies for enhancing humanization skills.

Thanks for pointing out these enhancements. As we previously stated, a more in-depth explanation of how the metaverse improves humanization abilities and which are the related virtual reality technologies has been included in Discussion section, as follows:

In lines 371-375 (pages 10-11) different possibilities of virtual reality are presented, including in Metaverse related to soft skills education.

In lines 402-407 (page 11) different applications of Metaverse in medical practice are presented.

In lines 409- 418 (page 11) as the field of education could also benefit from Metaverse-related technology.

  1. Despite the fact that this is a "Systematic Review Paper," there is still a requirement to highlight an architectural flow on how Metaverse-related technologies can improve humanization skills.

Thanks a lot for providing us this consideration. However, the application of the Metaverse in educational and training processes is in its enfant stage, so it is early to clearly imagine an architecture that shows us all the application possibilities for improving humanization skills.

Taking this into account, we added some references regarding Videogames (lines 371-375, pages 10-11), AR (Augmented Reality) and VR (Virtual Reality (lines 413-418, page 11)

  1. When you refer to "humanization," there is neither a conceptual nor a practical description of what you mean by this term. I mean there is no concept or operational definition of what you mean by Humanization.

Thanks for your recommendation. In this new version, an extensive explanation on humanization has been included, both in the introduction (lines 52-71, page 2) and in the discussion (lines 342-359, page 10).

  1. SECTION 4, which is supposedly where you should have outlined your review inference, but unfortunately you left it empty; if you had broadly explained "how to find a balance between obliging empathy and persuasive empathy" in another concept, that would have been a great contribution to the work; unfortunately, you did not do so.

At this point, thank you again for the reviewer's suggestion about the interest that a more in-depth analysis of the obliging empathy and persuasive empathy concepts would give to the discussion. It has certainly been a great contribution for the manuscript improvement, as follows:

  • Lines 455-529 (pages 12-14): the difference between obliging empathy and persuasive empathy is presented, as well as an extensive reflection on the development environment that gives rise to the empathic response and the correlation with the moral involvement of the caregiver that allows us to talk about a new concept: responsible empathy.

Best regards,

Authors.

Reviewer 2 Report

This paper is well organized and written. With a few more modifications, the quality of this paper can be improved significantly. The following concerns should be addressed before the acceptance:
1. The authors are suggested to add a new section to discuss some preliminaries of metaverse-related technologies.

2. The authors are suggested to add some diagrams in Section 4 to make a clearer understanding for readers.

3. The authors can discuss some scenarios of metaverse-related in healthcare sector.

4. The authors are suggested to add a new section to discuss the research directions of metaverse-related technologies in the healthcare environment.

Author Response

Reviewer 2

We do really appreciate your kind report about our manuscript.

Thanks for all your comment, which helped us to improve the manuscript.

Please see below the point-by-point response to each comment suggested.

This paper is well organized and written. With a few more modifications, the quality of this paper can be improved significantly. The following concerns should be addressed before the acceptance:

  1. The authors are suggested to add a new section to discuss some preliminaries of metaverse-related technologies.

Following the suggestion, we have included in the Introduction section an explanation about the technology of the Metaverse, as follows:

lines 72-76 (page 2) explain the changes undergone in education because of the incorporation of Information and Communication Technologies (ICT), which have promoted new methods in the different teaching-learning processes with special attention to the academic growth processes in healthcare students.

In lines 77-83 (page 2) we refer to the proliferation of extended reality, both augmented reality and virtual reality, and their capacity to create and add information from a real environment.

In lines 84-93 (page 2) we define Metaverse and what categories it presents and the characteristics that make it a training and learning tool in the context of education.

  1. The authors are suggested to add some diagrams in Section 4 to make a clearer understanding for readers.

Thank you very much for helping us to improve the manuscript, as well as to make the text more understandable and readable. We have added a Figure 2 (page 10, line 360), and a diagram (Figure 3) in Section 4, between the lines 483 and 486 (pages 12-13).

  1. The authors can discuss some scenarios of metaverse-related in healthcare sector.
  2. The authors are suggested to add a new section to discuss the research directions of metaverse-related technologies in the healthcare environment.

Thanks for your recommendation. Suggestions 3 and 4 have been taken up in a more detailed explanation of point 4. A discussion has been included on how the metaverse enhances humanization capabilities and what are the virtual reality technologies related to the healthcare practice, as follows:

In lines 371-375 (pages 10-11) different possibilities of virtual reality are presented, including in Metaverse related to soft skills education.

In lines 402-411 (page 11) different applications of Metaverse in medical practice are presented.

In lines 413- 418 (page 11) as the field of education could also benefit from Metaverse-related technology.

Thanks again and best regards,

Authors.

Round 2

Reviewer 1 Report

The format that has been modified is a significant step forward in terms of quality, however while the revised structure is a substantial improvement in terms of quality, it lacks a detailed explanation of figure 2, which is essential because of the need to clarify its components in detail and the associated investigations that led to the concept's first conceptualization. Other than that, the updated format is a huge improvement.

Author Response

Dear reviewer.

We do really appreciate your kind report.

All the information provided in Figure 2 is extensively explained in Discusion section. It corresponds to the advantages and disadvantages of Metaverse-related technologies explained in Discusion. 

However, we have modified the Figure 2´ title (page 16, line 593), as exactly the same refered as second subheading in Discusion section.

We hope it will help to clarify the Figure 2 content.

Thank you very much for taking the time to revise this manuscript, and consequently, helping us for its significative improvement.

Best regards,

Authors.

Reviewer 2 Report

The authors have incorporated all my concerns. There are no further requirements from my side.

Author Response

Dear reviewer.

We do really appreciate your kind report. Thank you so much for taking the time to revise this manuscript, and consequently, helping us for its significative improvement.

Best regards,

Authors.